# Nrp1 is Activated by Konjac Ceramide Binding-Induced Structural Rigidification of the a1a2 Domain

**DOI:** 10.3390/cells9020517

**Published:** 2020-02-24

**Authors:** Seigo Usuki, Yoshiaki Yasutake, Noriko Tamura, Tomohiro Tamura, Kunikazu Tanji, Takashi Saitoh, Yuta Murai, Daisuke Mikami, Kohei Yuyama, Kenji Monde, Katsuyuki Mukai, Yasuyuki Igarashi

**Affiliations:** 1Lipid Biofunction Section, Faculty of Advanced Life Science, Hokkaido University, Sapporo 001-0021, Hokkaido, Japan; dmikami@sci.hokudai.ac.jp (D.M.); kyuyama@pharm.hokudai.ac.jp (K.Y.); kt_mukai@jp.daicel.com (K.M.); yigarash@pharm.hokudai.ac.jp (Y.I.); 2Bioproduction Research Institute, National Institute of Advanced Industrial Science and Technology (AIST), Sapporo 062-8517, Hokkaido, Japan; y-yasutake@aist.go.jp (Y.Y.); n-tamura@aist.go.jp (N.T.); t-tamura@aist.go.jp (T.T.); 3Computational Bio Big-Data Open Innovation Laboratory (CBBD-OIL), AIST, Tokyo 169-8555, Japan; 4Department of Neuropathology, Institute of Brain Science, Hirosaki University Graduate School of Medicine, Hirosaki 036-8562, Aomori, Japan; kunikazu@hirosaki-u.ac.jp; 5Department of Medicinal Chemistry, Faculty of Pharmaceutical Sciences, Hokkaido University of Science, Sapporo 006-8585, Hokkaido, Japan; saito-t@hus.ac.jp; 6Faculty of Advanced Life Science, Hokkaido University, Sapporo 001-0021, Hokkaido, Japan; ymurai@sci.hokudai.ac.jp (Y.M.); kmonde@sci.hokudai.ac.jp (K.M.); 7R&D Headquarters, Daicel Corporation, Tokyo 108-8230, Japan

**Keywords:** ceramide, konjac, semaphorin3A, neurite outgrowth, neuropilin 1, endoglycoceramidase, sphingadienine

## Abstract

Konjac ceramide (kCer) is a plant-type ceramide composed of various long-chain bases and α-hydroxyl fatty acids. The presence of d4t,8t-sphingadienine is essential for semaphorin 3A (Sema3A)-like activity. Herein, we examined the three neuropilin 1 (Nrp1) domains (a1a2, b1b2, or c), and found that a1a2 binds to d4t,8t-kCer and possesses Sema3A-like activity. kCer binds to Nrp1 with a weak affinity of μM dissociation constant (Kd). We wondered whether bovine serum albumin could influence the ligand–receptor interaction that a1a2 has with a single high affinity binding site for kCer (Kd in nM range). In the present study we demonstrated the influence of bovine serum albumin. Thermal denaturation indicates that the a1a2 domain may include intrinsically disordered region (IDR)-like flexibility. A potential interaction site on the a1 module was explored by molecular docking, which revealed a possible Nrp1 activation mechanism, in which kCer binds to Site A close to the Sema3A-binding region of the a1a2 domain. The a1 module then accesses a2 as the IDR-like flexibility becomes ordered via kCer-induced protein rigidity of a1a2. This induces intramolecular interaction between a1 and a2 through a slight change in protein secondary structure.

## 1. Introduction

Semaphorin 3A (Sema3A) is a secretory signaling protein, belonging to the large semaphorin family, that acts as a guidance signal for growth cone collapse and nerve repulsion in axonal outgrowth [1], and also as a guidance signal for intracellular repulsion for axonal trafficking [2] through activation of Sema3A signaling, via the Sema3A receptor neuropilin 1 (Nrp1). Beyond the nervous system, Sema3A is also associated with various physiological and pathophysiological processes, such as cell migration, immune responses, angiogenesis, and cancer [3]. Sema3A is often up-regulated in some cancers [4,5,6] but down-regulated in others [7,8]. These conflicting effects indicate that Nrp1 may be a dual receptor for vascular endothelial growth factor (VEGF) and Sema3A. Sema3A may appear to be an ambiguous target for the development of cancer therapies. We previously identified a potential Nrp1 regulator called konjac ceramide [9], the characteristics of which suggest Sema3A-like agonism and antagonism [10].

Sema3A signaling works opposite to nerve growth factor (NGF) in the stratum corneum in heathy human skin, in which extracellular levels of Sema3A and NGF regulate skin barrier maintenance, preventing sensory nerves from extending neurites, which results in skin hypersensitivity [11,12]. 

In plants, free ceramides, which we refer to as ceramides herein, occur in lower abundance than glucosylceramide (GlcCer) [13] (Figure 1), which may have beneficial effects by preventing metabolic syndrome and related diseases [14,15]. Konjac (*Amorphophallus konjac* K.Koch) is a food plant rich in konjac (k) GlcCer that can prevent transepidermal water loss in mice and humans [16], and is used as a health food and in cosmetics. Konjac ceramide (kCer) is prepared by deglucosylation of kGlcCer using endoglycoceramidase I (EGCase I; Figure 1A), and has a neurite outgrowth inhibitory effect similar to that of Sema3A [17]. Previously, we demonstrated that kCer binds to Nrp1 and induces collapsin response mediator protein 2 (CRMP2) phosphorylation, leading to depolymerization of microtubules in the same way as Sema3A [9,10]. The functional part of kCer is the site of Nrp1 that binds Sema3A, and the specific active component of kCer (d4t,8t-kCer) involves 8-trans-isomerization (d18:2^4t,8t^) of the sphingadienine portion [18] (Figure 1B,C). Nrp1 is a co-receptor with Sema3A/Plexin A1 (PlexA1), which forms a signaling transduction system. Nrp1 is comprised of a large N-terminal modular ectodomain of approximately 850 amino acids, followed by a short membrane-spanning domain (24 residues) and a cytoplasmic domain (40 residues) [19]. 

The ectodomain is composed of five individual structural motifs or modules (a1, a2, b1, b2, and c; Figure 2A). Sema3A binds to the a1 module of the a1a2 domain through the Sema domain, the Ig-like domain, and the *C*-terminal basic tail, as well as to the b1 module through the *C*-terminal tail [1,20]. VEGF and heparin bind to the b1b2 domain (Figure 2B) [21]. We are interested in the molecular mechanism of kCer binding to Nrp1. In our previous study, the dissociation constant (Kd) of kCer indicated weak binding (μM) [10]. However, we wondered whether d4t,8t-kCer may bind to the a1a2 domain more tightly under certain conditions. Bovine serum albumin (BSA) has been used to solubilize kCer, due to the insolubility of ceramides. Herein, we tested two hypotheses (Figure 2C): (1) the low affinity (μM) of kCer for Nrp1 may be caused by a rapid degradation of the kCer/BSA/Nrp1 complex in the presence of BSA; and (2) the kCer –Nrp1 interaction may be of higher affinity (nM) than the kCer/BSA/Nrp1 complex (μM), but kCer–Nrp1 formation may be difficult due to the insolubility of kCer.

In the present study, we demonstrated that the a1a2 domain binds kCer, and an intrinsically disordered region (IDR)-like flexibility in a1a2 may be associated with kCer-mediated Nrp1 activation. In addition, an interaction site on a1a2 was explored by molecular docking, which revealed a possible Nrp1 activation mechanism.

## 2. Materials and Methods

### 2.1. Preparation of Chemicals

#### 2.1.1. Konjac Ceramide (kCer) 

Konjac Ceramide (kCer) was prepared in our laboratory according to a published procedure [17].

#### 2.1.2. Sphingadienine, d4t,8t-, and 4t,8c-C16Cer 

Sphingoid bases were prepared from kGlcCer, as described previously [22]. Then 4-trans,8-cis sphingadienine (d18:2^4t,8c^) and 4-trans,8-trans-sphingadienine (d18:2^4t8t^) were isolated by ODS-HPLC as described previously [23]. To condense sphingoid bases and palmitic acid, sphingadienine and palmitic anhydride were incubated in methanol at 37 °C for 22 h. The resulting ceramides were purified with an ODS-4 column (particle size: 5 µm; diameter: 10 mm; length: 150 mm; GL Science, Tokyo, Japan). Ceramide-containing materials were identified by infusion electrospray ionization-tandem mass spectrometry (ESI-MS/MS) analysis using a TripleTOF 5600 system (AB SCIEX, Foster City, United States). 

#### 2.1.3. NBD-ceramides (NBD-Cers) 

Sphingadienine (3.2 mg; 10.7 μmol) and 6-((7-nitrobenzo[c](1,2,5)oxadiazol-4-yl)amino)hexanoic acid (4.1 mg, 13.9 μmol; NBD-dodecanoic acid; Abcam, Cambridge, United Kingdom) were dissolved in *N,N*-diisopropylethylamine (200 μL) and dimethylformamide (DMF; 2 mL). The reaction mixture was cooled to 0 °C and stirred for 30 min at the same temperature. Then 1-[Bis(dimethylamino)methylene]-1H-benzotriazolium-3-oxide hexafluorophosphate (5.3 mg; 13.9 μmol) was added to the mixture and warmed to room temperature, and then stirred for 6 h. The reaction solvent was removed, and the crude residue was diluted with ethyl acetate (EtOAc). The organic layer was washed with 1 M HCl and brine, and then dried over MgSO_4_. After removing the solvent, the crude residue was purified by silica-gel column chromatography (EtOAc/*n*-hexane = 1:2, and then EtOAc alone) to yield NBD-sphingadienine ceramide (5 mg, 82%) as a yellow viscous oil.

Sphingosine and phytosphingosine were obtained from Avanti polar lipids (Alabaster, AL, United States). NBD-ceramides (NBD-Cers) were prepared as described above. NBD-Cers (d4t,8t-NBD-Cer, d4t,8c-NBD-Cer, d4t-NBD-Cer, phyto-NBD-Cer) were used for binding assays (Appendix A).

#### 2.1.4. Rhod-Bovine Serum Albumin (Fatty Acid-Free)

BSA (fatty acid-free; Sigma-Aldrich, St. Louis, United States) was labeled using a Rhodamine Fast Conjugation Kit (ab188286; Abcam, Cambridge, United Kingdom). According to the technical manual, the ratio of relative fluorescence units for Rhod per BSA molecule was calculated as 55,100 FRU/nmol.

### 2.2. Preparation of Recombinant Proteins

Recombinant, alkaline, phosphatase-fused Sema3A (AP-Sema3A) protein was prepared using HEK293 cells transiently transfected with an AP-Sema3A plasmid (#29448, chick origin) from Addgene (Watertown, United States) using a Lipofection Kit (ScreenFecT A plus; Wako Co., Osaka, Japan). The procedure was performed as described previously [24].

Alkaline phosphatase (AP) activity was determined using a LabAssay ALP activity assay kit (Takara, Shiga, Japan) with *p*-nitrophenylphosphate (pNPP; Sigma-Aldrich) as a substrate.

The gene encoding Nrp1 a1a2 (residues 23−269) was amplified from Nrp1–Fc–His (plasmid #72097; Addgene, Watertown, MA, USA), and the gene fragment was cloned into the *Nde*I and *Xho*I restriction enzyme sites of the pCold-ProS2 vector (Takara, Shiga, Japan) to incorporate a His6-tag at the N-terminus. Nrp1 a1a2 was expressed in *Escherichia (E.) coli* BL21(DE3) at 15 °C for 24 h by inducing with isopropyl-β-D-thiogalactoside (IPTG) at a final concentration of 0.1 mM. Cells were harvested and lysed by sonication in Buffer A (50 mM Na phosphate pH 8.0, 300 mM NaCl, 10% glycerol). The crude extract was added to Ni-NTA resin (Sigma), and the sample was eluted with a linear gradient of 0−400 mM imidazole in Buffer B (50 mM Na phosphate pH 6.0, 300 mM NaCl, 10% glycerol). Pooled fractions were dialyzed against Buffer C (50 mM TRIS-HCl pH 8.0, 10% glycerol) and further purified using DEAE-Sepharose FF resin with a linear gradient of 0−600 mM NaCl in Buffer C. Pooled fractions were dialyzed against Buffer D (10 mM K phosphate pH 7.0, 10% glycerol) and further purified using hydroxyapatite resin with a linear gradient of 10−400 mM K phosphate containing 10% glycerol. Purified a1a2 was dialyzed against Buffer E (25 mM TRIS-HCl pH 7.5, 10% glycerol) and concentrated to ~1.0 mg/mL. Genes encoding Nrp1 b1b2 (residues 270−588) and Nrp1 c (residues 589−854) were also amplified from Nrp1–Fc–His, and gene fragments were cloned into the *Nde*I and *Xho*I sites of the pET22 vector to incorporate a His6-tag at the C-terminus. Both Nrp1 b1b2 and Nrp1 c domains were expressed in *E. coli* BL21 (DE3) cells at 25 °C for 16 h with induction by IPTG at a final concentration of 0.1 mM. After harvesting, cell lysates were prepared by sonication in Buffer A, and His-tagged proteins were purified using Ni-NTA resin with a linear gradient of 0−400 mM imidazole in Buffer B. Pooled fractions were dialyzed against Buffer C and further purified using DEAE-Sepharose FF resin. Samples were eluted with a linear gradient of 0−400 mM NaCl in Buffer C. Purified b1b2 and c were dialyzed against Buffer F (25 mM TRIS-HCl pH 8.0, 10% glycerol) and concentrated to ~1.0 mg/mL. 

### 2.3. Preparation of Gene-Silencing Cells

Gene silencing was performed using small interfering RNA (siRNA). Plexin A1-specific siRNA was purchased from Life Technologies Japan Ltd. (4427037), with the sense strand sequence 5′-CCAAAGGAGUCAGCACUGUtt-3′. To compare the efficiency of Plexin A1 knockdown, scrambled siRNA (medium GC Duplex; Invitrogen, Grand Island, United States) was used as a negative control. Transfection of siRNA was carried out using Lipofectamine 2000 (Invitrogen), according to the manufacturer’s instructions. Briefly, siRNA and Lipofectamine 2000 reagent were mixed in Opti-MEM (Invitrogen) and incubated for 5 min at room temperature to allow complexation.

### 2.4. Co-Immunoprecipitation (Co-IP) of Sema3A and Nrp1 Domain Proteins

To examine which domain of Nrp1 binds kCer, His-tagged a1a2, b1b2, and c domains (10 pmol) were separately co-immunoprecipitated in 100 μL phosphate-buffered saline (PBS) buffer by 2 μg anti-6x-His monoclonal antibody (anti-His mAb; Thermo Fisher Scientific, Waltham, United States) and 10 pmol AP-Sema3A (0.25 μmol/min/mg protein). Before Co-IP, some ceramide samples (10 μM kCer or C18Cer) were mixed with and without a His-tagged domain to explore the involvement of AP-sema3A in His-tagged protein/anti-His mAb complex formation. AP activity was measured by a LabAssay ALP AP activity assay kit with p-nitrophenylphosphate as substrate (Wako Co., Osaka, Japan). C18Cer was purchased from Avanti polar lipids.

### 2.5. Quantitative Dot Blot Analysis

To examine the binding affinity of kCer to Nrp1 domain proteins, NBD-Cers were tested against each domain (a1a2, b1b2, and c) by dot-blotting with a Bio-Dot SF Microfiltration Apparatus (Bio-Rad, Berkeley, CA, United States). Mixtures of 0.05% Tween-20 and 50 mM TRIS-buffered saline (TBST) containing d4t,8t-NBD-Cer (2, 20, 50, 70, 100, 200, 600, 800, 1200, and 1600 nM; 50,000 FRU/nmol) and a1a2 (100, 200, 600 nM) were dotted onto a nitrocellulose membrane (0.45 μm, #162-0117; Bio-Rad, Berkeley, CA, USA), aspirated, and washed three times with TBST buffer. The wet membrane was quickly sealed in a transparent sheet, and fluorescence intensity was quantitated at an excitation wavelength of 473 nm and an emission wavelength of 520 nm, using a Typhoon FLA7000 instrument (GE Healthcare, Tokyo, Japan). Binding parameters (Kd and Bmax) were obtained using a best-curve fitting method performed by nonlinear regression analysis using the GraphPad Prism 5.0 software package (GraphPad, San Diego, United States). 

### 2.6. Time Course of Fluorescence Intensity of NBD-Cer and Rhod-BSA on the Cell Surface 

PlexinA1 gene-silencing PC12 cells were treated with d4t,8t-NBDCer (100 nM, 50,000 FRU/nmol) and Rhod-BSA (100 nM, 55,100 FRU/nmol) at 4 °C for 30 min in Dulbecco’s modified Eagle’s medium (DMEM) containing 0.1% fetal calf serum, rinsed with cold PBS three times, and incubated in cold DMEM for 0, 5, 15, and 30 min. Subsequently, cells were rinsed again with cold PBS three times and fixed with 5% glutaraldehyde/PBS at room temperature for 20 min. The fluorescence intensity for NBD and Rhod was analyzed at excitation and emission wavelengths of 473 and 520 nm and 530 and 556 nm, respectively, using a FLUOVIEW FV10i instrument (Olympus, Tokyo, Japan). Green and red fluorescence was analyzed by ImageJ and Fiji, respectively. Luminance and colocalization ratio were subsequently determined.

### 2.7. Differential Scanning Calorimetry (DSC)

Differential scanning calorimetry (DSC) thermograms were collected using a VP-capillary DSC platform (GE Health Care, Tokyo, Japan) at temperatures up to 90 °C and scan rates of 0.2 or 1.0 °C/min [25]. For measurements, protein concentrations in the presence and absence of 100 μM ceramides were adjusted to 3 mg/mL in Dulbecco’s phosphate buffered-saline (DPBS) buffer pH 7.2 (14287-080; Thermo Fisher Scientific, Waltham, MA, United States) containing 100 mg/L CaCl_2_, 100 mg/L MgCl_2_, 1 g/L glucose, and 36 mg/L sodium pyruvate. Ceramides C24-, C18-, C16-, C8-, and C2-Cer were purchased from Avanti polar lipids (Albaster, AL, USA).

### 2.8. Circular Dichroism (CD) Spectroscopy

Circular dichroism (CD) spectra of 5 μM a1a2 and 5 μM a1a2 plus 1 μM kCer were analyzed using a JASCO J-820 spectrometer at 20 °C [26]. Both were dissolved in DPBS buffer. CD data were measured from 190 to 260 nm at a scanning rate of 100 nm/min with a single scan. CD data are presented in terms of ellipticity (θ degrees), normalized by scaling to molar concentrations of the repeating unit, where molar ellipticity is calculated using the following formula:[*θ*] = 100 x *θ_obs_* / (*c* x *N* x *l*)(1)
where *θ_obs_* is the observed ellipticity in degrees, *c* is the molar concentration of protein, *N* is the number of amino acids in the protein, and *l* is the cell path length in cm. The secondary structure composition based on CD spectra was analyzed by the JASCO multivariate Secondary Structure Estimation (SSE) program.

### 2.9. Molecular Docking

The atomic model of the Nrp1 a1 domain (residues 25−141) was obtained from the X-ray structure of Nrp1 a1–a2–b1–b2 (PDB code 4gz9) [27]. Atomic models for d4t,8t-C16kCer, and d4t,8c-C16kCer (C16h:0) were built and refined using MF MyPresto version 3.2 (FiatLux). Molecular docking between Nrp1 a1 and d4t,8t-C16kCer/d4t,8c-C16kCer was performed using Autodock Vina version 1.1.2 [28]. The configured input files for docking calculations were generated by Autodock Tools version 1.5.6 [29]. Nrp1 a1 was treated as a rigid body, and a docking box that fully contains the whole model of the a1 domain was set up with a grid dimension of 38 × 38 × 38 Å. Three independent docking simulations were performed with an exhaustiveness parameter of 8, 80, and 800. A total of 60 docking poses, consisting of the top 20 docking poses for each run, were obtained and superimposed on the structure of the a1 domain. Molecular figures were prepared using PyMOL version 2.3.4 [30]. 

### 2.10. Statistical Analysis

The number (*n*) in each experimental condition is shown in the figure legends. Data were analyzed statistically using Prism 5.0 (GraphPad). For comparison of two experimental conditions, statistical analysis was performed using a paired *t*-test. Alternatively, statistical analysis was carried out using one-way analysis of variance (ANOVA), followed by Tukey’s multiple comparison post-tests and Dunnett’s tests. A *p-*value <0.01 was considered significant. * indicates significantly different results. Ranges of *p*-values are indicated in the figure legends.

### 2.11. Research Rules and Regulation

All experiments were performed with approval from the regulatory boards of Hokkaido University.

## 3. Results

### 3.1. Co-Immunoprecipitation of AP-Sema3A and Nrp1 Domain Proteins

AP-Sema3A was quantified in 0, 50, and 100 nM AP-Sema3A/100 nM, His-tagged protein/150 nM, anti-His mAb (2 μg) complexes (Figure 3A). AP-Sema3A in Co-IP samples was increased to 2−4 pmol for Nrp1 and 3−5.5 pmol for a1a2. However, AP-sema3A was not co-immunoprecipitated with b1b2 or c. The addition of kCer revealed decreased involvement of AP-Sema3A in Co-IP of Nrp1 and a1a2, but C18Cer did not induce any change in AP-Sema3A (Figure 3B). Neither kCer nor C18Cer induced an increase or decrease in AP-Sema3A for any of the Nrp1 domain proteins.

### 3.2. Dot-Blotting and Binding Parameters

Binding between NBD-Cer (d4t, 8t-, d4t, 8c-, phyto-, d4t-) and Nrp1 domains (a1a2, b1b2, c) was analyzed by dot-blotting (Figure 4A). Clear binding to d4t,8t-NBDCer was only observed for a1a2. The other NBD-Cer derivatives did not exhibit any binding. Saturation binding for a1a2 was observed using a serial dilution of d4t,8t-NBDCer. In the absence of a1a2, non-specific binding of the ligand to membranes was observed in a dose-dependent manner (Figure 4B). 

Non-specific binding was eliminated by the addition of $\sqrt{C*X}$ to the binding equation (Figure 4B). Nonlinear regression analysis of curve fitting yielded a dissociation constant *K_d_* of 114 ± 5 nM (*n* = 3), and saturated binding values for 100 nM a1a2 gave a *B_max_* value of 58000 ± 8000 FI/nmol (*n* = 3). To examine the number of binding sites in the a1a2 protein, 100, 200, and 600 nM a1a2 was used for a serial dilution of d4t,8t-NBD-Cer (Appendix A). There was a linear relationship between a1a2 and saturated d4t,8t-NBD-Cer, with a gradient of 1.038 (Appendix A). Thus, the a1a2 domain displayed nanomolar binding to d4t,8t-kCer via a single binding site, as predicted in hypothesis 1 (Figure 2C).

### 3.3. Dissociation Time Course for Cell Surface Binding of NBD-Cer and Rhod-BSA

To determine whether the kCer–BSA interaction with the a1a2 domain of Nrp1 is accompanied by a higher dissociation than kCer alone, NBD-Cer and Rhod-BSA were mixed and added to the PC12 cell culture. Using the time course shown in Figure 5A, the dissociation and association of NBD-Cer and Rho-BSA were compared. The d4t,8t-NBD-Cer remained present in cells even after 30 min, but Rhod-BSA disappeared faster from cells (Figure 5B). However, d4t,8c-NBD-Cer disappeared faster from cells, similar to Rhod-BSA (Figure 5C). Phyto- and d4t-NBDCer also disappeared at the same rate as Rhod-BSA (data not shown). 

Next, we examined whether the longer duration of d4t,8t-NBD-Cer in cells may be due to binding to the a1a2 domain of Nrp1. Nrp1 domain proteins were mixed with d4t,8t-NBD-Cer/Rhod-BSA, and cells were subsequently treated. Immediate washing and fixing of cells indicated that domains may substitute upon binding between the NBD-Cer/Rhod-BSA complex and Nrp1 on the cell surface. As shown in Figure 6A, a1a2 inhibited d4t,8t-NBD-Cer–Rhod-BSA formation, but not formation of the d4t,8c-NBDCer/Rhod-BSA complex. By contrast, b1b2 and c did not alter the interaction between NBD-Cers and Rhod-BSA.

In addition, we showed that d4t,8t-NBD-Cer followed a different time course for colocalization ratio than Rhod-BSA, while the time course for d4t,8c-NBD-Cer was the same as that of Rhod-BSA (Figure 6B). We demonstrated that binding of d4t,8t-kCer/BSA to Nrp1 yields an unstable complex that rapidly releases BSA, resulting in a stable d4t,8t-kCer-Nrp1 complex consistent with hypothesis 2 (Figure 2C).

### 3.4. Comparison of Differential Scanning Calorimetry Thermograms and Circular Dichroism Spectra Between Control and kCer

To examine potential interactions between kCer and BSA, the unfolding of BSA by kCer was compared with other ceramides and kGlcCer using DSC measurements. Table 1 shows changes in phase transition temperature (Tm) for BSA following the addition of ceramide samples. kCer increased Tm to 63.1 °C compared with the control (61.4 °C). Animal-type ceramides (C24Cer, C18Cer, C16Cer) had less influence on Tm, but artificial ceramides (C8Cer and C2Cer) increased Tm similarly to kCer. However, Tm elevation by kGlcCer could be reversed by blocking a hydroxyl of kCer by glucosylation. Thus, the hydroxyl of kCer is important for interaction with BSA.

The a1a2 domain is structurally unstable, based on the lack of a Tm peak during heating denaturation (Appendix A). The broad peak indicates that partial unfolding may occur during thermal denaturation. By contrast, the addition of kCer resulted in the appearance of a denaturation peak at 52.5 °C. This suggests that kCer may promote folding of the partially unfolded structure of a1a2. CD spectral changes are shown in the ΔCD spectrum (Appendix A), and Table 2 shows changes in the secondary structure of a1a2. There was a small change in the secondary structure, with an increase in α-helical content (1.4%) and a decrease in β-turn content (1.7%). This small change may be due to thermal denaturation at a relatively low temperature.

### 3.5. Molecular Docking Simulation

To obtain structural insight into kCer binding to Nrp1, we performed in silico molecular docking studies. The a1 module is believed to bind Sema3A; hence, the structure of a1 was based on the previously reported a1a2b1b2 structure [27], and was used as a docking target for kCer (d4t,8t-C16kCer). Because the kCer binding site is unclear, a large simulation box containing the overall structure of the a1 module was constructed to search for possible kCer binding regions throughout the a1 model, and three independent docking simulations were performed with varying exhaustiveness parameters. Three regions of the a1 molecular surface (Site A, B, and C) were identified as potentially suitable for kCer binding; in addition, 70% of the docking poses (42/60) are concentrated at Site A, a shallow groove between the side chains of Lys26 and Tyr38 (Figure 7A). It should also be noted that site A is located in close proximity to the Sema3A binding region containing Pro73 and His74, identified as essential residues for Sema3A binding in previous work [27] (Figure 7A).

## 4. Discussion

Ceramides are composed of sphingosine and fatty acids, and frequently contain sphingolipids such as glycosphingolipids, sphingomyelins, and free ceramides on the mammalian cell membrane [31]. Animal-type ceramides can act as signaling molecules to regulate cellular functions, such as apoptosis [32,33], and as signaling mediators in other processes [34], resulting in epidermal barrier formation [35] and homeostatic control of brain function [36]. We previously studied a plant-type ceramide that regulates Nrp1 [9,10,17,18,24], with potential as a novel cancer therapeutic agent.

In the present study, we analyzed Sema3A-like activity by investigating the kCer binding site on Nrp1. In the first step of Sema3A signaling, at the cell surface, Sema3A forms a hetero-pentameric complex with a dimer of its receptor Nrp1, and this complex recruits a dimer of plexin A1 (PlexA), resulting in a hetero-pentameric complex that possesses signal-transducing activity [37]. This receptor complex activates CRMP2 phosphorylation downstream of Sema3A signaling, following which pCRMP2 no longer binds to tubulin microtubules, leading to their destabilization and depolymerization, and subsequent collapse of the growth cone [38]. 

Previously, we showed that kCer is a potential Sema3A-like ligand of the extracellular region of Nrp1, but animal-type ceramides do not exhibit Sema3A-like activity [9,10]. Nrp1 is a cell surface receptor for the semaphorin III family and VEGF ligands, and binding results in keratinocyte migration [39] and transendothelial migration of lymphocytes [40]. The long-chain base structure of plant-type ceramides (Figure 1B) is more diverse than that of animal-type ceramides. kCer, present in most plant-type ceramides, is comprised of various long-chain bases and fatty acids (Figure 1C). In a previous study, we investigated which side chain groups of kCer are essential for Sema3A-like activity and Nrp1 binding activity, and found that an α-hydroxyl group is not needed on the fatty acid, and the length of the fatty acyl chain is not critical; meanwhile, 8-trans-isomerization of sphingadienine (d18:2^4t, 8t^) is essential for activation, but 8-cis isomerization (d18:2^4t,8c^) is not effective [18].

In the present study, we confirmed that AP-Sema3A is bound only by the a1a2 region of the extracellular domain proteins (a1a2, b1b2, and c). Unlike C18Cer, kCer displayed specific binding activity and was competitive with AP-Sema3A (Figure 3A,B). The interaction between kCer and the a1a2 domain is species-specific, hence d4t,8t-kCer is active but d4t,8c is not. Dot-blotting experiments with NBD derivatives of kCer and d4t,8t-kCer yielded a nanomolar dissociation constant (*Kd* = 113.8 nM) and a single binding site on a1a2 (Appendix A), consistent with hypothesis 1 (Figure 2C).

After being distributed in the circulating blood system, drugs and lipids bind to serum proteins [41]. This binding is reversible, and a dynamic equilibrium exists between bound and unbound molecules. It is believed that only unbound molecules can exert their pharmacological effects by penetrating through cell membranes into cells. Among serum proteins, lipoprotein-carrying transport system components are important for lipids, such as cholesterol and triglycerides. By contrast, serum albumin plays a particularly important role in drug binding and in distributing drugs and external components, such as chemically modified lipids [42,43]. Our DSC experiments (Table 1) revealed a stronger interaction between kCer and BSA than occurs with typical animal-type ceramides, such as C24Cer, C18Cer, and C16Cer. BSA is a useful carrier protein for kCer and artificial ceramides like C2Cer and C8Cer. This structural interaction is lost following glucosylation of kCer, suggesting that the hydrophilic region of kCer is important for binding to BSA. In an extracellular milieu such as DMEM, the kCer/BSA complex is accessible to cultured PC12 cells, while kCer/BSA is thought to bind more weakly than the kCer/Nrp1 complex. Thus, kCer may be transferred to Nrp1 via the kCer/BSA/Nrp1 complex, which is transient and unstable, resulting in kCer/Nrp1 complex formation. We tested this proposal (hypothesis 2 in Figure 2C). Dissociation time course analysis of NBD-Cer and Rhod-BSA from the cell surface of PC12 cells revealed that d4t,8t-kCer is bound to the cell surface through the a1a2 of the extracellular domains of Nrp1 (Figure 5 and Figure 6), and BSA is rapidly removed from the d4t,8t-kCer/BSA/Nrp1 complex on the cell surface. 

Thermal denaturation of a1a2 did not occur (Appendix A), but kCer altered the structural stability of a1a2. IDRs have unique structural and functional characteristics [44]. They lack well-defined three-dimensional structures in the absence of ligand binding, and are considered to behave as random coils [45]. Our CD spectra indicated secondary structural changes in the presence and absence of kCer (Appendix A), consistent with an IDR-like character. There was a slight but appreciable change in α-helical, β-sheet, and β-turn content (Table 2), consistent with changes in intramolecular interactions in the a1a2 domain. 

Molecular docking studies identified three distinct kCer (d4t,8t-C16kCer) binding sites (Site A, B, and C) on the molecular surface of a1 (Figure 7A), near the Sema3A binding site in a1. These observations imply that binding of kCer to Site A might cause similar changes in the structure of Nrp1 to those caused by binding of Sema3A, although the exact details of the structural effects of kCer and Sema3A still need to be elucidated. We also performed docking simulations using d4t,8c-C16kCer, but there were no significant differences in docking convergence between 8-trans and 8-cis isomerization of sphingadienine. The docking results did not provide a basis for discriminating between the mechanisms of d4t,8t-C16kCer and d4t,8c-C16kCer; rather, they simply identified the molecular surface region (Site A) that most likely binds kCer. In addition, the hydroxyl groups of most d4t,8t-C16kCer docking poses in Site A interact with main-chain amino acids of a1. As shown in Appendix A, two hydroxyls of kCer typically form hydrogen bonds with main-chain carbonyl oxygen and amide nitrogen atoms of Glu34, Asn35, and Pro36. Consistent with the DSC results for BSA in the presence of kCer (Table 1), hydrogen bonding interactions between the hydrophilic portion of kCer and amino acids of a1 appear to be important in the molecular docking simulation (Figure 7A and Appendix A). 

Based on our results, we propose a possible mechanism for the activation of Nrp1 by kCer (Appendix A). The a1 module is far away from a2 and b1b2, and it may therefore possess IDR-like flexibility. When kCer binds to Site A of the a1 module, this flexibility may diminish as the IDR-like region becomes more rigid, strengthening interactions between a1and a2, resulting in a slight change in secondary structure.

## Figures and Tables

**Figure 1 cells-09-00517-f001:**
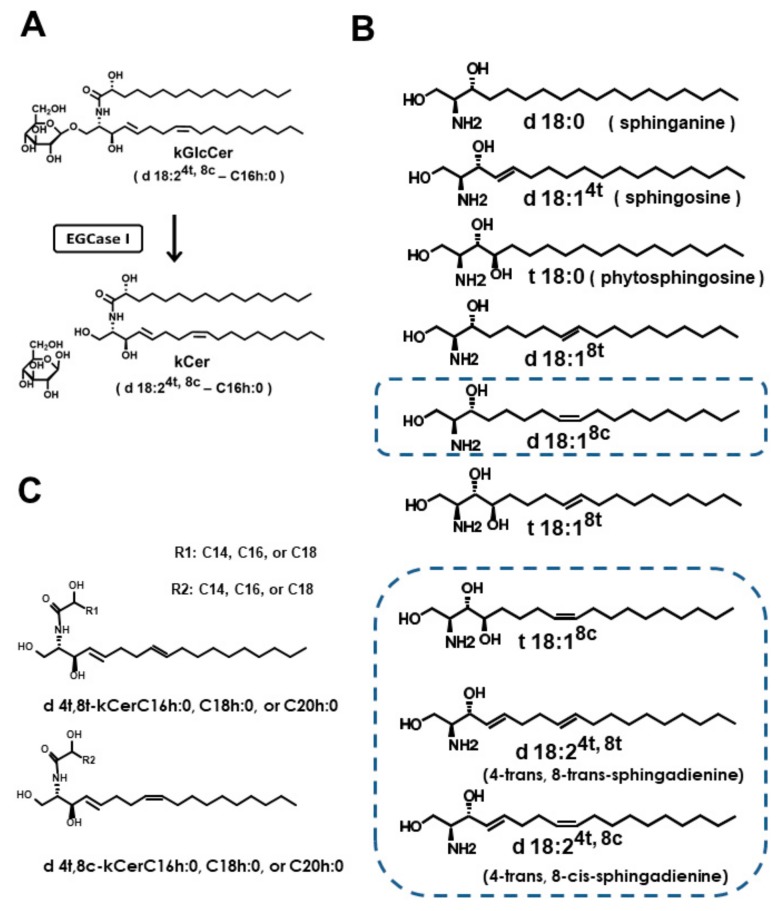
The nomenclature of long-chain bases and ceramides following the recommendations of the IUPAC-IUBMB Joint Commission. (**A**) The endoglycoceramidase (EGCase) reaction of konjac glucosylceramide (kGlcCer). Plant-type ceramides can be prepared from plant-type glucosylceramide (GlcCer) by EGCase I treatment. A major molecular species (d18:2^4t, 8c^-C16h:0) of konjac ceramide (kCer) is shown. (**B**) The main long-chain bases found in plants. The long-chain bases of kCer produced by EGCase treatment are delineated by a dotted line rectangle. (**C**) kCer molecular species generated by EGCase treatment of kGlcCer. The length of the carbon chain of each hydroxyl fatty acid (C16 to C20) is shown.

**Figure 2 cells-09-00517-f002:**
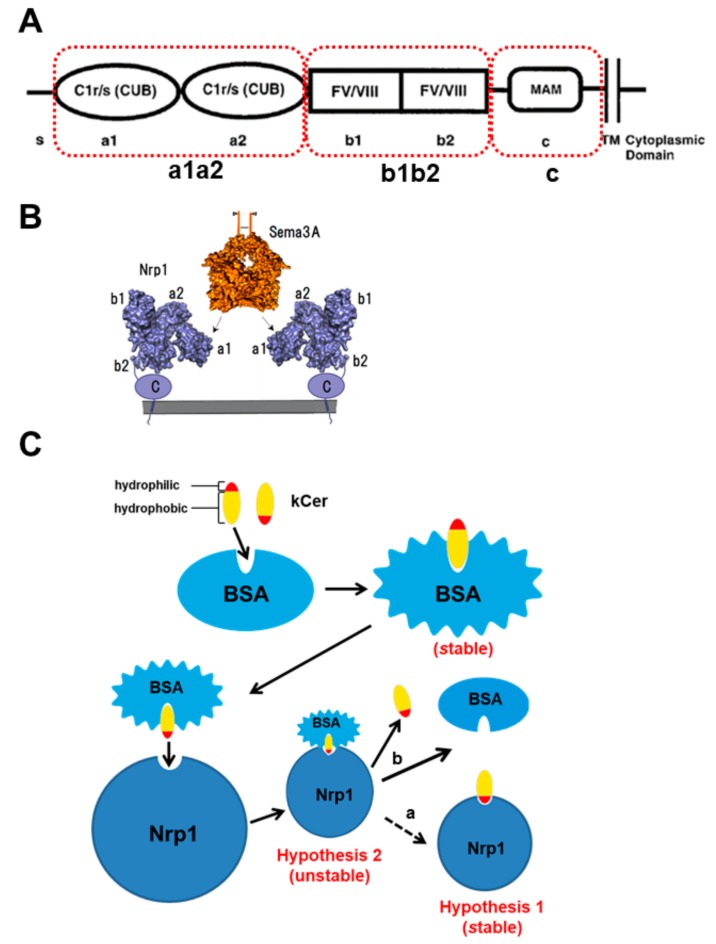
Structure of Neuropilin 1 (Nrp1) and the binding mechanism of semaphorin 3A (Sema3A). (**A**) Diagram displaying the modular structure of Nrp1, which is comprised of five domains (a1, a2, b1, b2, and c), as illustrated by He et al. [1]. S: signal peptide; C1r/s: complement (CUB); FV/VIII: regions homologous to coagulation factor V and VIII; MAM: a specific domain in transmembrane proteins. (**B**) Interaction between Sema3A and Nrp1. Sema3A binds to domain a1 of a1a2 via the Sema domain, the Ig-like domain, and the *C*-terminal basic tail, as well as to domain b1 via the *C*-terminal tail. Vascular endothelial growth factor (VEGF) and heparin bind to b1b2. (**C**) Proposed model for kCer binding to Nrp1 as a Sema3A agonist based on the two hypotheses. A hydrophobic part of kCer binds to the fatty acid binding pocket, forming the kCer/BSA complex, which releases kCer via formation of the kCer/BSA/Nrp1 complex. kCer is indicated by a dashed arrow (a), representing a weak flow relative to the bold arrow (b).

**Figure 3 cells-09-00517-f003:**
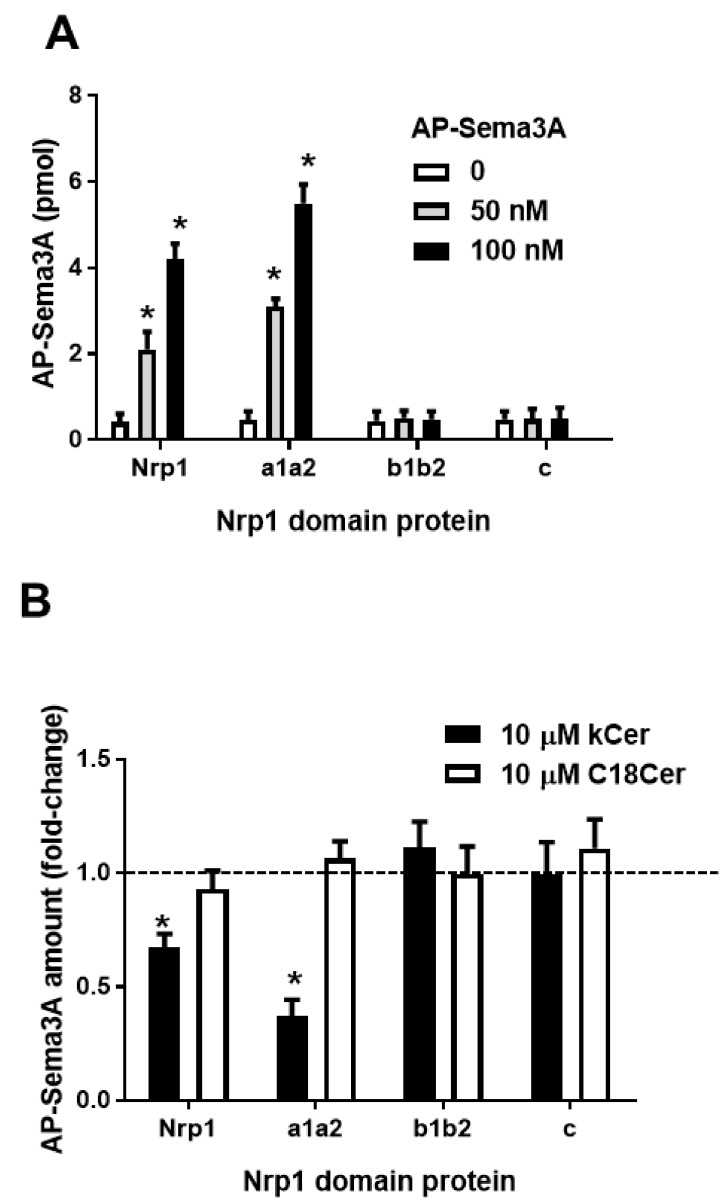
Co-immunoprecipitation (Co-IP) of alkaline, phosphatase-fused Sema3A (AP-Sema3A) using the His-tagged Nrp1 domain and anti-His antibody. (**A**) AP-Sema3A (0, 50, or 100 nM) was mixed with Nrp1, a1a2, b1b2, or c (100 nM) prior to Co-IP with an anti-6x-His monoclonal antibody (anti-His mAb) (2 μg). (**B**) kCer or C18Cer (100 μM) and AP-Sema3A (100 nM) were mixed with Nrp1, a1a2, b1b2, or c (100 nM) before Co-IP with anti-His mAb (2 μg).

**Figure 4 cells-09-00517-f004:**
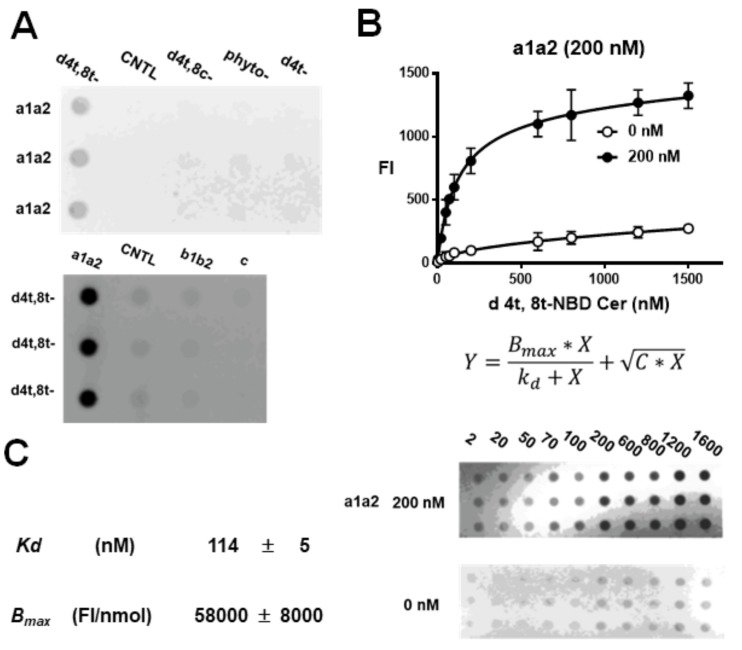
Dot blot analysis of the binding characteristics of kCer to Nrp1 domain proteins. (**A**) The upper blot shows the results for mixtures a1a2 (of 100 nM) plus d4t,8t-NBD-Ceramide (-NBD-Cer), d4t,8c-NBD-Cer, phyto-NBD-Cer, or d 4t-NBD-Cer (all at 100 nM). The lower blot shows the results for mixtures of 100 nM d4t,8t-NBD-Cer with 100 nM a1a2, b1b2, or b. (**B**) Saturation curve of d4t,8t-NBD-Cer binding to 200 nM a1a2 (●). Control blot of d4t,8t-NBD-Cer without a1a2 is shown on the lower plot (○). Data are presented as means ± standard deviation (SD) (*n* = 3). *X* is the concentration of d4t,8t-NBD-Cer (nM). The *y*-axis label “FI” represents the fluorescence intensity of d4t,8t-NBD-Cer bound to 200 nM a1a2. *C* is a constant for non-specific binding of d4t,8t-NBDCer. Non-specific binding is represented in the equation as $\sqrt{\mathrm{CX}}$. (**C**) *Kd* is the dissociation constant of the binding of d4t,8t-NBD-Cer to a1a2. *B_max_* is the plateau of the binding of d4t,8t-NBD-Cer to a1a2. FI is the fluorescence intensity per nanomole a1a2.

**Figure 5 cells-09-00517-f005:**
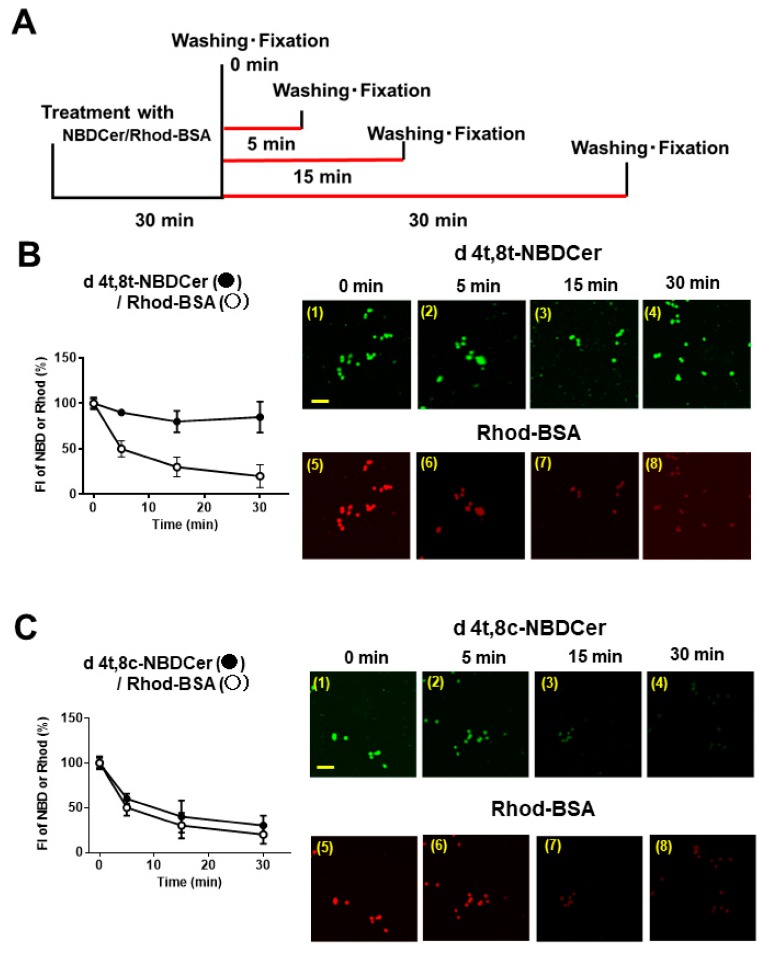
Time course of NBD-Cer and Rhod-bovine serum albumin (BSA) bound to PC12 cells. (**A**) Dissociation time course analysis of FI based on binding of 100 nM d NBD-Cer (FI = 5000) and 100 nM Rhod-BSA (FI = 5510), examined using Plexin A1 gene-silencing PC12 cells. (**B**) Images showing (1 to 4) changes in d4t,8t-NBD-Cer and (5 to 8) changes in Rhod-BSA at the indicated timepoints. The left graph shows a time course plot of FI (%) relative to 0 min. Data are presented as mean ± SD (*n* = 3). Scale bar = 100 μm. (**C**) Images showing (1 to 4) changes in d4t,8c-NBD-Cer and (5 to 8) changes in Rhod-BSA at the indicated timepoints. The left graph is a time course plot of FI (%) relative to 0 min. Data are presented as mean ± SD (*n* = 3). Scale bar = 100 μm.

**Figure 6 cells-09-00517-f006:**
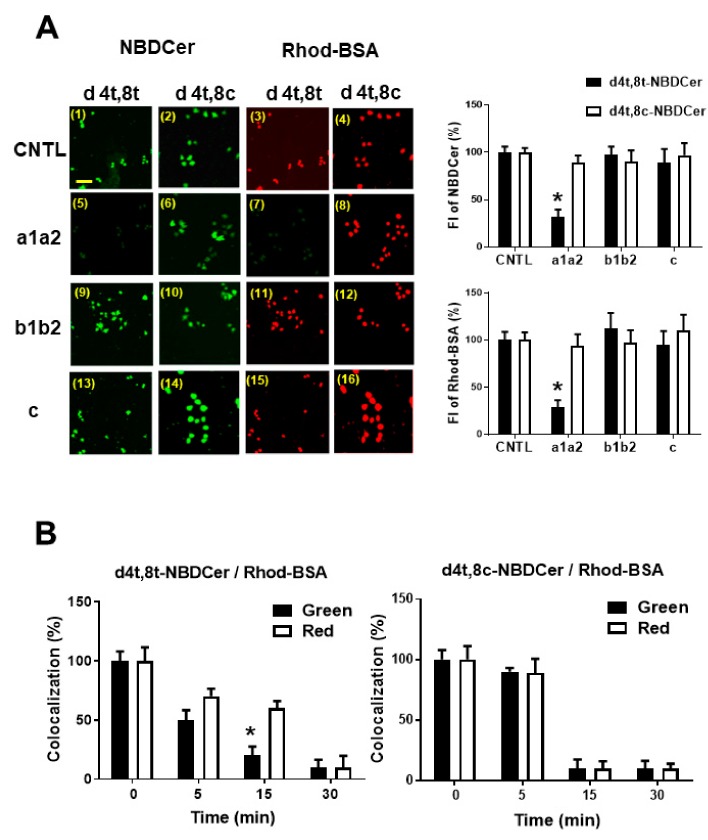
Inhibition profile of the Nrp1 domain of NBD-Cer and Rhod-BSA binding to PC12 cells. (**A**) Representative images of Plex A1 gene-silencing PC12 cells examined together with Nrp1 domains a1a2, bib2, and c (100 nM), or control (CNTL) plus 100 nM NBD-Cer/100 nM Rhod-BSA. Scale bar = 100 μm. Changes in FI are shown in the right upper graph (NBD-Cer) and lower right graph (Rhod-BSA). Data are presented as mean ± SD (*n* = 3; **p* < 0.01). (**B**) Colocalization of d4t,8t- or d4t,8c-NBD-Cer and Rhod-BSA during dissociation (0 to 15 min) in cells. Green (■) indicates occupation based on NBD fluorescence, and Red (□) indicates occupation based on Rhod fluorescence. Data are expressed as FI (%) at 0 min and are presented as mean ± SD (*n* = 3; **p* < 0.01).

**Figure 7 cells-09-00517-f007:**
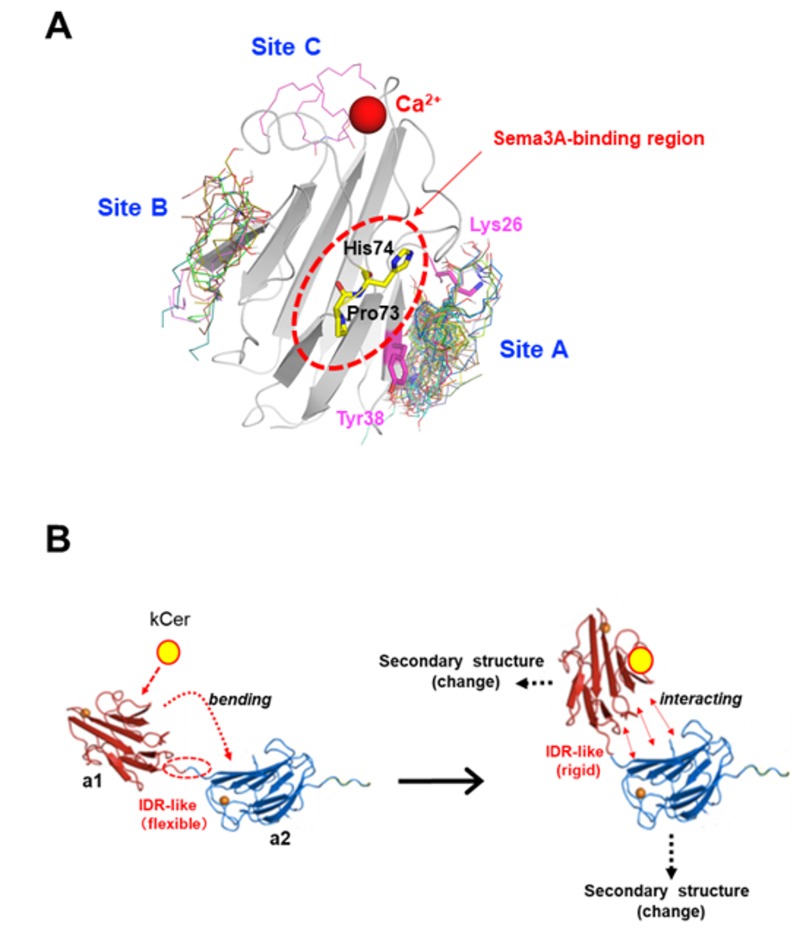
(**A**) Molecular docking simulation for the a1 module and d4t,8t-C16kCer, and artificial species of kCer composed of C16:0 fatty acid and d4t,8t-sphingadienine. There are three binding sites (A, B, and C) on the a1 protein. Site A is located near the Sema3A binding region. (**B**) Possible activation mechanism of Nrp1 by kCer. The sole a1 module is far away from a2 and b1b2 (Appendix A). intrinsically disordered region (IDR)-like flexibility likely occurs due to the distance between a1 and a2 modules. When kCer binds to Site A of the a1 module, the IDR-like flexibility between a1 and a2 diminishes, and the IDR-like region rigidifies, strengthening the intermolecular interactions between a1 and a2, resulting in a slight change in protein secondary structure.

**Table 1 cells-09-00517-t001:** Differential scanning calorimetry (DSC) analysis of potential interactions between BSA (3 mg/mL) and ceramides (100 μM).

CNTL	kCer	kGlcCer	C24Cer	C18Cer	C16Cer	C8Cer	C2Cer
61.4 °C	63.1 °C	61.0 °C	61.9 °C	61.9 °C	61.9 °C	62.5 °C	64.1 °C

CNTL: control, no ceramide; ceramides: addition of kCer, kGlcCer, C24-, C18-, C16-, C8-, or C2Cer.

**Table 2 cells-09-00517-t002:** Secondary structure of a1a2 (0.03 mg/mL) in the presence of kCer (1 μM).

Common element/Additive	α-helices	β-sheets	β-turns	Random Coil	Total
CNTL	15.5%	43.4%	5.2%	35.0%	100.0±0.6%
kCer	16.9%	44.8%	3.5%	34.8%	100.0±0.4%

CNTL: control, no kCer; kCer: addition of kCer.

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
