# Peer review of "Nrp1 is Activated by Konjac Ceramide Binding-Induced Structural Rigidification of the a1a2 Domain"

_cells, 2020, doi:10.3390/cells9020517_

Round 1
Reviewer 1 Report
In this manuscript, Usuki et al aim to understand how kCer binds and activates Nrp1 to provide a molecular basis for the design of potential disease treatment regimen targeting this interaction. The authors report that AP-Sema3A is bound only by the a1a2 region of the extracellular domain proteins and unlike C18Cer, kCer displayed specific binding activity and was competitive with AP-Sema3A. Using a series of biochemical and biophysical approaches, the authors propose a new mechanism for the activation of Nrp1 by kCer. Overall the experiments are designed to support the key conclusions, and the manuscript provides an advanced understanding for kCer interaction with Nrp1. The reviewer finds the manuscript is kind of difficult to follow and suggest seeking for help from English expert to further polish the writing. In addition, there are some concerns that need to addressed before consideration for publication.
Fig 3: purity of these proteins used should be indicated. One concern is if proteins purified from HEK293 cells will have contaminations from binding proteins.
Fig 4A: resolution for this image needs to improve.
Fig 5B-5C: downstream effects for d 4t, 8c-NBDCer addition into PC12 cells would need to be examined.
Fig 7: great work on the docking- if the authors can validate this model by point mutants in a1 module it will be convincing.
Author Response
We appreciate very much the interest of the editor and reviewers in reviewing this manuscript. The reviewers’ comments and suggestions are perceptive, fair, and constructive, which are important to improving the presentation of this work. We have addressed each of their comments, and believe that the revised version can meet the journal’s publication requirements.
Responses to comments from Reviewer 1
- English expert to further polish the writing.
Response: The revised manuscript has been checked by English proofreading service.
- Fig 3: purity of these proteins used should be indicated. One concern is if proteins purified from HEK293 cells will have contaminations from binding proteins.
Response: The reviewer’s concern is very right. The co-immunoprecipitation in Fig. 3 might be influenced by the contaminations. However, the binding characteristics of these proteins were also confirmed by dot blot analysis (Fig. 4A). We think the purity has not changed on the results. Just in case, we add Fig. S5 of supplemental material and show the recombinant protein purity and contamination.
- Fig 4A: resolution for this image needs to improve.
Response: Fig 4A is replaced by an improved image.
- Fig 5B-5C: downstream effects for d 4t, 8c-NBDCer addition into PC12 cells would need to be examined.
Response: The downstream of Sema3A signaling has been blocked by silencing PlexinA1, that is a coupling receptor with Nrp1, while Sema 3A binds Nrp1 on the cell surface. In Fig 5B and C, the experiment was aimed on the receptor-ligand binding on the cell surface. We think it is out of the present study to examine whether or not d 4t, 8t- or d 4t,8c-NBDCer might show any influence on the downstream pathway.
In addition, NBD-modified lipids might behave as alien substance on mammalian cells that is targeted by drug-resistant ABC transporters. If we test the downstream effect by d 4t, 8t- or d 4t,8c-NBDCer, the outcome might be complicated to analyze contribution of only Sema3A pathway.
- Fig 7: great work on the docking- if the authors can validate this model by point mutants in a1 module it will be convincing.
Response: We sincerely thank the reviewer for his/her critical comment. We fully agree with reviewer that the point mutants in a1 is important for validation of the presented docking models. We think that the exhaustive mutational analysis for the residues creating Site A/B would be required for complete understanding of the kCer binding site in a1, which will be planned as another research in the future.

Reviewer 2 Report
The authors presented studies carried out with kCer - plant ceramide. They quantified binding affinities of kCer towards neuropilin 1 – Nrp1 domains and described BSA effect on the interactions.
The manuscript is written well and prepared properly. I kindly suggest several points for the authors to consider to improve the text and presentation of their results.
Abstract: page 1 line 29 - Instead ”…interaction. and found that…“ should be “interaction. We found that…“ Abstract: page 1 line 36-8 -The last clause starting: “Given the Sema3A-mediated…” should be omitted, as it is misleading considering the results presented within the manuscript. Results: page 9 line 278 and all other appearances in the text and Figure 4 - measured values should be reported rounded to one significant digit e.g. 114 ± 5, 58 000 ± 8000. Results: Consider abbreviating d4t,8t-NDBCer, and d4t,8c-NDBCer to simpler and clearer short names e.g. tCer, and cCer that would improve readers’ orientation. Results: page 12 line 323 – “3.4 DSC and CD” should be changed to a more descriptive paragraph title. Results: page 12 line 341 Table 2 – Standard deviations should be included to determine the significance of the changes objectively. Discussion: page 15 lines 442-5 - The last paragraph should be removed in accordance with the suggestion 2 mentioned above.
Author Response
Responses to comments from Reviewer 2
- Abstract: page 1 line 29 - Instead ”…interaction. and found that…“ should be “interaction.
Response: According to the reviewer’s comment, the sentence has been changed. In page 1 line 30.
- We found that…“ Abstract: page 1 line 36-8 -The last clause starting: “Given the Sema3A-mediated…” should be omitted, as it is misleading considering the results presented within the manuscript.
Response: The sentence “Given the Sema3A-mediated….” Is removed from the Abstract.
- Results: page 9 line 278 and all other appearances in the text and Figure 4 - measured values should be reported rounded to one significant digit e.g. 114 ± 5, 58 000 ± 8000.
Response: I agreed with the indication of one significant digit in the binding constants, because I think it is not high- precision to calculate the parameter values by fitting the fluorescence intensity plot using the binding equation. The measured values are changed in Fig. 4 and page 9 line 286.
- Results: Consider abbreviating d4t,8t-NDBCer, and d4t,8c-NDBCer to simpler and clearer short names e.g. tCer, and cCer that would improve readers’ orientation.
Response: Diversity of long chain bases is important information to understand our study.
Therefore, we have already shown the nomenclature of long-chain bases in Fig. 1 and the abbreviations: page 16 line 479 and 480. We used this abbreviation in our previously published articles. We are anxious about that readers might have some of confusion for different abbreviations and difficulty to get an entirely understanding.
- Results: page 12 line 323 – “3.4 DSC and CD” should be changed to a more descriptive paragraph title.
In page 12 line 329, the paragraph title, “3.4 DSC and CD” has been changed to a new one “Comparison of DSC thermograms and CD spectra between CNTL and kCer”.
- Results: page 12 line 341 Table 2 – Standard deviations should be included to determine the significance of the changes objectively.
Response: Standard deviation of regression analysis is added to the Table 2.
- Discussion: page 15 lines 442-5 - The last paragraph should be removed in accordance with the suggestion 2 mentioned above.
Response: According to the reviewer’s comment, the last paragraph is removed from the Discussion.

Round 2
Reviewer 1 Report
The authors largely argued with my previous concerns- one figure panel was replaced. I am fine with others but I do think it is important to validate the structure simulation model by point mutations and a quick binding assay.